# Education and Awareness on Antimicrobial Resistance in the WHO African Region: A Systematic Review

**DOI:** 10.3390/antibiotics12111613

**Published:** 2023-11-10

**Authors:** Walter Fuller, Otridah Kapona, Aaron Oladipo Aboderin, Adeyemi Temitayo Adeyemo, Oluwadamilare Isaiah Olatunbosun, Laetitia Gahimbare, Yahaya Ali Ahmed

**Affiliations:** 1World Health Organization Regional Office for Africa, Brazzaville P.O. Box 06, Congo; gahimbarel@who.int (L.G.); aliahmedy@who.int (Y.A.A.); 2Zambia National Public Health Institute, Lusaka 10101, Zambia; kaponaotridah@gmail.com; 3Department of Medical Microbiology & Parasitology, Obafemi Awolowo University Teaching Hospital Complex, Ile-Ife 220005, Nigeria; diipo_aboderin@yahoo.com (A.O.A.); adeyemiadeyemo3@gmail.com (A.T.A.); dare.olat@yahoo.com (O.I.O.)

**Keywords:** awareness, antimicrobial resistance, antimicrobial stewardship, education, one health, systematic review

## Abstract

This review documents the status of AMR education and awareness in the WHO African region, as well as specific initiatives by its member states in implementing education and awareness interventions, as a strategic objective of the Global Action Plan on AMR, i.e., improve knowledge and understanding on AMR through effective communication, education, and training. A systematic search was conducted in Google Scholar, PubMed, and African Journals Online Library according to Preferred Reporting Items for Systematic Reviews and Meta-analyses (PRISMA) guidelines, for articles published in English. Retrieval and screening of articles was performed using a structured search protocol following a pre-set inclusion/exclusion criterion. Eighty-five published articles reporting 92 different studies from 19 Member States met inclusion criteria and were included in the final qualitative synthesis. Nigeria (21) and Ethiopia (16) had most of the studies, while the rest were distributed across the remaining 17 Member States. The majority of the articles were on knowledge, attitude, and practices with regard to AMR and antimicrobial use and most of them documented a general lack and suboptimal knowledge, poor attitude and practices, and widespread self-medication. This review shows low levels of knowledge of AMR coupled with extensive misuse of antimicrobial medicines by different target audiences. These findings underscore the urgent need for enhanced and context-specific educational and positive behavioural change interventions.

## 1. Introduction

Antimicrobial resistance (AMR) has been acknowledged as one of the top ten public health threats facing humanity [1]. It is a complex and multidimensional problem, threatening not only human and animal health, but also regional, national, and global security, and the economy. O’Neil’s 2014 report projected that AMR will cause 10 million deaths annually by 2050 and will result in between 2% to 3.5% less in gross domestic product (GDP), if not adequately and urgently addressed [2]. The report by the Global Research on Antimicrobial Resistance (GRAM) project estimated that 4.95 million deaths were associated globally with bacterial resistance in 2019 and, of these, 1.27 million deaths were directly attributable to resistance [3]. The consequences of AMR are more calamitous in low to middle-income countries (LMICs) and their development agenda due to the occurrence of more prevalent infectious diseases.

In response to the threat of AMR in the context of the One Health approach, in 2015 the World Health Organization (WHO) endorsed a Global Action Plan (GAP) on AMR that serves as a blueprint for country-specific National Action Plans (NAPs) [4]. One of the objectives highlighted in both the GAP and country NAPs is the need to improve awareness and understanding of antimicrobial resistance through effective communication, education, and training, the umbrella for the other four strategic objectives [4,5]. The GAP calls for among other things, increased national AMR awareness targeting different audiences/demographics in human health, animal health, and agricultural practices, establishing AMR as a core component of awareness campaigns, professional education, training, certification, and development across sectors, as well as the inclusion of antimicrobial use (AMU) and resistance in school curricula to promote better understanding and awareness [4].

Antimicrobial use and misuse across sectors and the spread of resistant pathogens and resistance determinants within and between sectors have been cited as the major drivers of AMR [6,7]. Antimicrobial use is largely influenced by knowledge, expectations, nature of practices, interactions of prescribers and patients, economic incentives, characteristics of the health system, and the regulatory environment [8]. This is largely because there still remains a major gap in implementing education and awareness interventions on judicious use of antimicrobials for the different target audiences in the region [9,10]. Countries have mostly confined their education and awareness activities to global commemorations, particularly, the World AMR Awareness Week (WAAW), which is usually characterized by small-scale AMR campaigns scattered around different sectors. Considering the complexity of AMR and its cross-cutting as well as multisectoral dimensions, episodic interventions are inadequate to impact or trigger sustainable behavioural change. We sought to document the level of awareness and understanding of AMR in the WHO African Region as well as the deployment of effective communication, education, and training.

## 2. Materials and Methods

### 2.1. Data Sources and Search Strategy

A systematic search was carried out in Google Scholar, PubMed, and African Journals Online Library according to Preferred Reporting Items for Systematic Reviews and Meta-analyses (PRISMA) guidelines. Published articles on AMR awareness and understanding documented from original studies conducted in the 47 countries of the WHO African Region were obtained. The search was carried out using relevant keywords which were pooled together using the Boolean term “OR”. The keywords were combined to form a final search strategy using the Boolean term “AND” as shown in Table 1. There was no restriction on the year of publication.

### 2.2. Selection Criteria

The criteria for inclusion of the articles in the qualitative synthesis were: The articles must be original and published in English, they should document findings related to AMR awareness, understanding, education, communication, and training in countries within the WHO African Region, and must be available in full text.

### 2.3. Article Selection Process

The initial 729,727 records from the databases were screened according to the relevance of the title. Of the 165 remaining articles, 44 duplicates (articles from more than one database) were removed. The retained articles (121) were further screened for inclusion by reading through the abstract and noting the presence of one or more of the keywords. The final screening was based on the inclusion criteria stated above and the articles that did not meet all the criteria were excluded from the final synthesis. Figure 1 shows the stepwise process of selection of the 85 articles for qualitative synthesis.

### 2.4. Data Extraction/Qualitative Synthesis

Relevant information synthesized from each article included the country where the study was conducted and the year of publication, the type of research tools and how they were deployed (questionnaires, self-administered or interviewer administered; physical or online/web-based), study settings (urban, semi-urban or rural), study participants (general public, students, healthcare workers, outpatients/hospitalized patients, veterinarians, farmers, etc.), number of study participants, as well as key findings including knowledge, understanding and awareness levels, trainings, and other relevant findings.

## 3. Results

The 85 articles available for qualitative synthesis reported 92 (80 single-country studies and five multi-country studies) original studies within the WHO African Region (Table 2). The selected studies were from 19 countries including Nigeria (21), Ethiopia (16), Tanzania (8), Zambia (8), Ghana (7), South Africa (7), Uganda (5), and others, as shown in Figure 2.

The majority of the country surveys were among human healthcare practitioners (32, 34.8%) and undergraduate healthcare students (21, 22.8%). The country surveys also involved general public/community dwellers (11), animal producers/farmers (10), patients (5), veterinary drug retailers and dispensers (4), community drug retailers (2), and veterinarians and veterinary paraprofessionals (2). Three other studies surveyed more than one group of respondents.

Human healthcare professionals (HCPs) generally have good knowledge of AMR as a global and national problem; it is, however, less appreciated as an issue in their local institutions or wards in day-to-day practice. Many HCPs perceived AMR as a problem for individuals who misuse antibiotics. Most HCPs identified drivers of AMR as overuse and misuse, fewer of the professionals identified other factors promoting the emergence and spread of AMR such as lack of antibiograms, lack of antibiotic treatment guidelines, poor hospital hygiene and infection prevention and control measures, and lack of awareness campaigns and education. Only one study conveyed that AMR can be promoted by poor access to antimicrobials. Many studies documented poor training on AMR among human HCPs, some reported low levels of formal training or up-to-date AMR information with the few available knowledge acquired as far back as during medical training. Some studies clearly reported requests by HCPs for more education on AMR; a study among Ethiopian medical interns reported that as high as 94% of respondents wanted more education on AMR. The review further shows that AMR knowledge levels/scores differed significantly across different professionals; it was generally better among physicians and pharmacists when compared to nurses and other HCPs (Table 2). In addition, two separate studies among dispensers in community retail outlets/patent medicine vendors showed overall good knowledge of AMR and its drivers, the knowledge however did not have an impact on their daily practice; the majority of them dispensed antimicrobials without prescription.

Six studies involved veterinary care workers and revealed that awareness and knowledge of AMR were higher among pharmacists than other veterinary drug dispensers. A third of all the dispensers in one of the studies knew that AMR could be caused by misuse. Good knowledge of veterinary pharmacists was associated with work experience of more than one year. A study of veterinary paraprofessionals in five community districts revealed that most did not attend refresher courses and seminars on AMR.

Twenty-six studies engaged university undergraduate students including 21 studies of human health care students (medical students—8; pharmacy students—3; medical laboratory science students—1; combined healthcare students—10), 3 studies of veterinary students, and 2 studies involving students in general. Pharmacy students and medical students had better knowledge of AMR, its drivers, and how AMR could be curtailed when compared with nursing students, medical laboratory science students, and other paramedical students. The knowledge level is generally poor among university students studying non-healthcare-related courses. In fact, one of the surveys involving 1320 participants showed that general university students displayed less knowledge than community dwellers that self-medication promotes AMR. The review also revealed that students at higher levels had better knowledge of AMR than those at lower levels. Most studies revealed a robust knowledge among healthcare students that AMR constitutes a global problem, many studies however revealed poor students’ knowledge that it is a local hospital problem affecting routine practice. Healthcare students in many studies did not also recognise that hand hygiene and IPC measures play important roles in the control of AMR. The review also showed a complete absence or inadequacy of training on AMR in healthcare.

Ten studies across the countries were among animal farmers among who were poultry farmers (5) and pastoralists/livestock farmers (3). Awareness and knowledge of AMR are generally poor and may vary across different country regions, and such awareness could be better among commercial than subsistent poultry farmers. The studies showed that many farmers did not perceive AMR as a public health issue, some perceived it as a problem of foreign countries, and some perceived it as a local problem that only affects individuals who regularly take antibiotics. The studies also showed that large numbers of farmers did not see antimicrobial use in animals as a promoter of AMR while many farmers did not have the knowledge that drug-resistant organisms can be transmitted from animals to humans.

The five studies among hospital patients (three in-patients and two out-patients) showed poor awareness and knowledge of AMR. The studies showed that a large number of patients neither know that inappropriate use of antimicrobials promotes AMR nor know that resistant infections prolong hospitalization and increase healthcare costs. A study, however, showed that the knowledge that AMR could affect mortality was good.

A review of the 11 studies which incorporated community dwellers and the general public showed a low level of awareness and education on AMR and poor knowledge of drivers of AMR. It further showed the possibilities of geographical variation in AMR knowledge. A study revealed that more than 50% of the public did not know that AMR is difficult to treat, and another study showed that more than 40% of the public did not know that AMR is more costly to treat. Less than one-tenth knew that hand washing can prevent transmission of AMR. Furthermore, knowledge of AMR was noted to be better among respondents with higher levels of education.

## 4. Discussion

Antimicrobial use is mostly influenced by knowledge, perception, prevailing attitudes, and practices on antimicrobials. Education and awareness play a critical role in addressing the use of antimicrobials and ultimately AMR, as it is an overarching objective that cuts across the other four objectives of both the global action plan and member state’s Action Plans for control of AMR. Expeditious and effective implementation of the One Health national plans to mitigate AMR requires an all-encompassing, robust, and society-wide education utilising target-specific, efficient communication strategies aimed at the government and policymakers, healthcare workers, veterinarians, animal farmers and food producers, community drug vendors, and high school, undergraduate, and graduate students, as well as the general public.

Despite a majority of countries (74%, 23/31) in the region holding regular public awareness campaigns against AMR and its drivers [95], our review of the literature highlighted poor levels of awareness and knowledge on AMR across societal strata. One of the key weapons against AMR is public awareness/engagement which, if conducted taking into account context-specific determinants and elements of behaviour change, has the potential of engendering behavioural change among the public bearing in mind that AMR is a society-wide issue demanding specified roles to be played by everyone [96]. The fact that the public is generally unaware of AMR and its dare consequences calls for aggressive orientation and community engagements among member countries with the full participation of the governments’ relevant sectors, civil society, non-governmental organisation, and the media for concerted and coordinated activities towards communicating AMR and its deleterious consequences to the populace in an effective manner for enhanced understanding. A previous report suggests that such community engagement, deploying context-driven community approaches as well as tools including appropriately packaged and positioned messaging, can go a long way to facilitate expected behavioural change in LMICs [97].

This review noted an inappreciable knowledge of AMR even among human HCWs; findings in a multi-country survey across Ghana, Nigeria, and Tanzania are particularly noteworthy showing that respondents with good awareness of AMR was not up to 60% in the majority of the countries and awareness scores differed significantly among the different professionals within each country [36]. This review also noted in two separate Nigerian studies that more than 40% of human HCWs even in urban centres do not have knowledge of AMR [63,66]. Our review findings therefore corroborate what was previously noted in a scoping review which documented global knowledge gaps on AMR in human health including in AMR burden and drivers as well as awareness and education, and with the African region leading the gap chart [98]. Even in studies where AMR knowledge levels were reported to be high among human healthcare professionals, high proportions neither know the extent of AMR nor its effect that AMR could lead to treatment failure [11,12]. In addition, some HCWs neither saw AMR as a problem in their local HCF nor appreciated its impact on their daily routine practice, with such knowledge dearth even documented among physicians in tertiary HCFs [15,37,43], with attendant serious consequences on healthcare cost and patient outcomes. Good knowledge of practices leading to overuse and misuse of antimicrobials driving AMR is noted among most HCWs; however, there are still widespread knowledge deficits on other key drivers such as lack of antibiogram as well as poor hand hygiene and IPC measures, two studies carried out in Ethiopia and Zambia [17,90] and another three conducted in Benin, Cameroun, and Uganda [12.15,83], respectively, documented knowledge gaps in the important roles of antibiogram and IPC measures in AMR. The review further showed in several studies the higher knowledge level of AMR among physicians and pharmacists compared with levels in other HCWs including nurses. It was also found in several studies that human HCWs had poor training on AMR; the majority had no current knowledge and a few claimed that the last training they had dated back to their student years. Our findings in this regard equally relate with those of a systematic governance analysis which reviewed the contents of NAPs on AMR from 114 countries and revealed that the scoring for education was about the lowest of the 18 domains [99], highlighting that basic and continuous education on AMR for health-care workers need to be robustly established in most countries. There is an important need among the member countries in the region for deliberate efforts and activities to institutionalise AMR education as part of workforce education in relevant sectors taking the form of pre-service training, or in-service training which demands champions and resources. In the human health sector, for example, countries can leverage the curricula guide provided by the World Health Organisation (WHO) for health workers’ education and training to develop a template for continuous professional development [100]. In addition, AMR learning should also be incorporated into high school and university curricula-based education.

Furthermore, the findings justify the use of unconventional approaches to systematizing education thereby promoting and ensuring sustainable behaviour change for addressing AMR threats. Countries within the region can leverage several educational initiatives targeted at different groups of people to improve awareness and education on AMR, for example, e-Bug Europe and MicroMundo for pre-university students, the “Do Bugs Need Drugs” program which is an initiative of Alberta Health Services and the British Columbia Centre for Disease Control, WHO-AFRO regional debate initiative which took place in Senegal during the 2022 continental World AMR Awareness Week celebration, the debate kit launched by of the Spanish National Plan for AMR, and the ReAct Campaign which educate on the nature as well as drivers of AMR. There are other smaller initiatives such as the “Bugs in Bangkok” board game, and the WHO-supported Dr. Ameyo Stella Adadevoh (DRASA) Health Trust model which provided a club way to teach secondary school students about AMR in Nigeria [101,102]. Likewise, the potential usefulness of social media for AMR awareness and education has not been appreciably explored in the region; in fact, the utility of social media for combating AMR has been regarded as a neglected approach especially in low- and middle-income countries [103]. It is therefore instructive that countries adapt and where possible deploy social media platforms for education and information dissemination, particularly among the youths in whom its use is highly rising and has a high tendency to bring about change in attitudes, practices, and perceptions and, ultimately, behaviour [104].

Farmers across the region showed abysmal levels of knowledge of AMR, many of them neither correlated the use of antimicrobials on their animals to AMR nor fathomed a possibility that resistant bugs could be transmitted from animals to humans. For example, a study from Ethiopia involving 571 rural farmers documented that only 41% knew that excessive use of antimicrobials in their animals can cause AMR. In the same vein, a Cameroon study of 358 farmers, among several others, reported a deficit in the knowledge that AMR could be transmitted from their animals to humans or the environment [13,27]. These findings have been well corroborated in a systematic review among poultry farmers which revealed that only 43% had knowledge about AMR and only about 50% understood the impacts of AMR on poultry, human health, and the environment [105]. The fact that knowledge of AMR is worse in LMICs had been earlier reported in a review of 103 multiregional study articles which showed that farmers in Africa and Asia demonstrated grossly deficient knowledge of AMR as compared with their counterparts from Europe [106]. The responsibility therefore rests on member countries to be committed to providing flexible methods to educate and inform community farmers about AMR which can be facilitated by existing platforms such as appropriate broadcast media for wide reach. Successes recorded by some countries can be replicated and upscaled by many other countries, a typical example is the poultry farmer field schools by FAO in Ghana and Kenya which facilitated a knowledge-driven reduction in antibiotic use in birds, improvement in IPC practices and enhancement of patronage with animal health professionals [107]. Furthermore, fortifying AMR education among veterinary and para-veterinary workers is a step in the right direction for the stepdown of robust knowledge to farmers. Being animal health practitioners and health extension workers, their education provides a direct impact on farmers’ knowledge of animal husbandry and drug resistance. Such education can be well guided at the level of the undergraduate and can utilise various platforms and tools including online-based resources deployed for enhanced education on antimicrobial resistance and antimicrobial use [108]. The region will do good by leveraging on previously used initiatives to strengthen veterinary training to incorporate AMR education [109].

## 5. Conclusions

To the best of our knowledge, this review is the first to document the status of AMR education and awareness in the WHO African Region. This review opens up the knowledge gaps in AMR across the board and sheds light on the need to design evidence-based, pragmatic, cost-effective, targeted interventions to improve public awareness and knowledge of AMR. The review reveals a need for improved educational strategies within member countries focusing on the inclusive academic curriculum at all levels of studies, continuous professional development, as well as utilising innovative and context-specific approaches for effective and sustained awareness and education on AMR.

## 6. Limitation

Articles included in this systematic review were restricted only to those published in English.

## Figures and Tables

**Figure 1 antibiotics-12-01613-f001:**
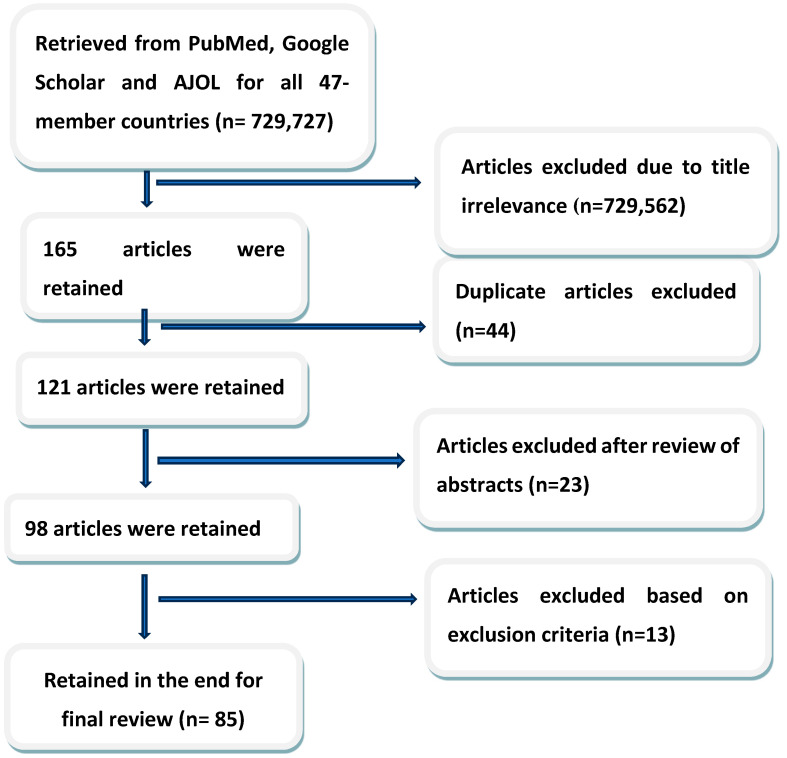
Diagram of the search and selection of review articles.

**Figure 2 antibiotics-12-01613-f002:**
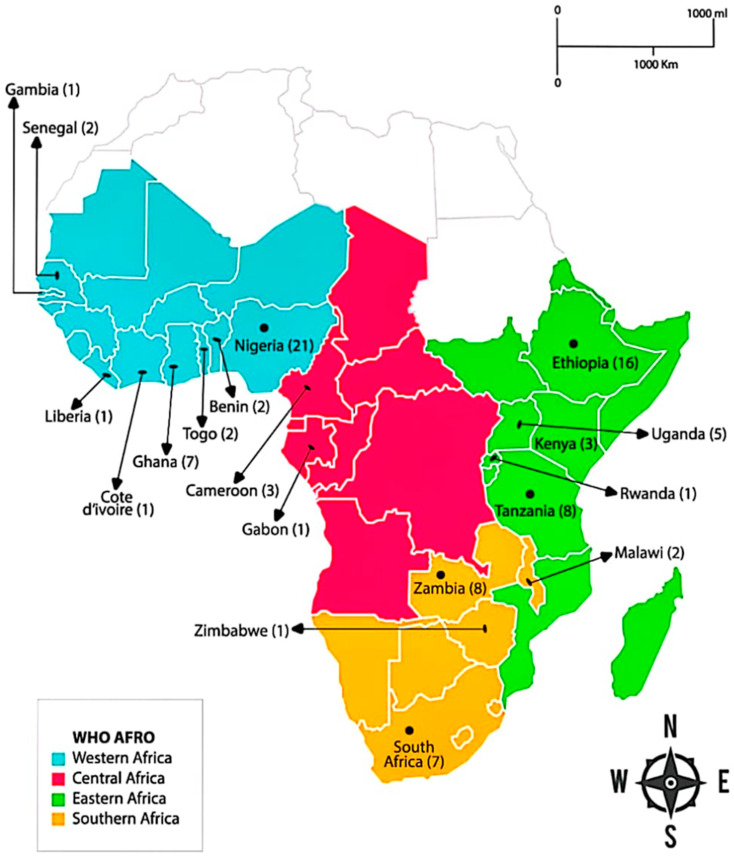
Map showing distribution of published articles on AMR awareness in WHO AFRO.

**Table 1 antibiotics-12-01613-t001:** Search Strategy.

“awareness OR understanding AND antimicrobial OR antibiotic AND resistance AND education OR communication OR training AND name of country”

**Table 2 antibiotics-12-01613-t002:** Summary of findings from qualitative synthesis of articles from studies across the countries in the WHO African Region.

S/N	Country	Study Period	Research Tool (Methodology, e.g., Self-Administered Questionnaire, Online Survey, Interview, Others)	Setting (Urban, Rural), Population—Human Healthcare Workers (Doctors/Nurses/Pharmacists), Patients, Animal Healthcare Workers (Farmers/Vets), General Public, Pre-Service Students: Medicine/Pharmacy/Nursing/Vet, Primary/High School, Others	Number (Subjects)	Survey Key Findings	Reference
1	Benin	2019	Self-administered questionnaire	Urban HCW (Prescribers: nurses, midwives, physicians, etc.)	330	-Most of the participants (70–84%) surveyed have a good knowledge (of antibiotic resistance but only 30–36% knew that AMR leads to treatment failure; 70–79% identified misuse as a root cause of AMR. Other causes less readily identified included poor antibiogram.	Dougnon et al., 2020 [11].
2	Benin	Aug–Dec 2018	Self-administered questionnaire	Urban Dispensers: different categories of Pharmacy Staff	159	-63.4% correctly defined AMR. -Causes of AMR identified as failure to comply with treatment duration (96.2%) and self-medication (94.9%). Other identified causes included SF (63.1%) and poor hygiene (16.6%). -71.7% had no idea of the current extent of resistance.	Allabi et al., 2023 [12].
3	Cameroon	Jun–Nov 2019	Interviewer-administered questionnaire	Rural Poultry farmers	358	-Low mean score knowledge of AMR with significant variation across regions (higher in some regions than others). -Risk perception including transmission from animals to humans, environment, and public health threat is grossly poor. -Did not elicit information about possible causes of AMR. -Level of education positively influences knowledge of AMR.	Moffo et al., 2020 [13].
4	Cameroon	Jan–Aug 2019	Self-administered questionnaire	Urban Antimicrobial Prescribers (100), dispensers (113), and users (385)	598	-92% of prescribers and 62.8% of dispensers could define AMR. -Multidrug resistance defined correctly by 72% and 34.5% of prescribers and dispensers, respectively. -90% and 78.8% of dispensers knew that AMR is a public health problem. -64% of prescribers knew that AMR is a multisectoral issue. AMR is a O-H issue by 90% of prescribers, 3.54% of dispensers, misuse drives resistance (89% prescribers, 20.35% dispensers), and no information by users.	Djuikoue et al., 2022 [14].
5	Cameroon	May–Feb 2019	Self-administered questionnaire (email)	Urban Physicians practising in tertiary care	98	-93% knew that AMR is a significant problem in the country, but 40% believed that AMR is a problem in their hospital wards. -54% of doctors disagreed that poor hand hygiene is a cause for spread of antibiotic-resistant bacteria.	Domche Ngongang et al., 2021 [15].
6	Cote d’Ivoire	Aug–Oct 2020	Survey including 2 countries in West Africa Self-administered (google forms, kobo)/interviewer-administered questionnaire	Urban Health Professionals (Physicians—79, Pharmacists—70, and Veterinarians—72)	221	-64% had good/very good knowledge of AMR. -Veterinarians had significantly higher knowledge of AMR than doctors and pharmacists (69% vs. 42% vs. 40%). -53% had no formal AMR training.	Bedekelabou et al., 2022 [16]^a^.
7	Ethiopia	Jun 2013	Self-administered questionnaire	Urban Physician—175, Nurses—210	385	-72.2% were knowledgeable about AMR. -Majority agreed that AMR is a global and national problem. -They identified poor adherence to antibiotics (86%) and overuse (80.5%) as leading causes of AMR. -Other causes are lack of local antibiogram (12.3%), self-prescription (53.5%), and poor awareness (9.2%).	Abera et al., 2014 [17].
8	Ethiopia	Jan–Mar 2018	Self-administered questionnaire	Urban prescribers in Veterinary drug retail outlets	108	-64.8% report AMR responsible for difficulty in treating infectious diseases. -60.2% know AMR is a global public health and economic threat. -Drivers or causes of AMR identified as the use of wrong antimicrobial (80.6%) or poor quality antimicrobial (79.6%) amongst others; 70.4% self-prescribe, 9 (8.3%) do not know causes of AMR, only 24 (22.2%) have had training on AMR.	Zeru et al., 2019 [18].
9	Ethiopia	Aug–oct 2019	Self-administered questionnaire	Urban Medical Interns in tertiary health facilities	270	-93.3% knew AMR as a national concern, in addition, 95.5% equally perceived it as an institutional problem. -Respondents had good knowledge of the drivers of AMR (90–95%). -94.8% would like more education on AMR.	Mersha, 2018 [19].
10	Ethiopia	Nov–Feb 2020	Self-administered/interviewer-administered questionnaire	Rural Farmer (farm owners/workers)	91	-90.1% have heard about AMR, 50% do not know about impact of AMR, 45% do not know about mode of transmission. -Respondents identified causes of AMR as: -76.9% of farmers agreed that AMR is caused by poor awareness. -Other causes of AMR noted by farmers include lack of rapid, and effective diagnostics (67%), substandard antibiotic use (64.8%), and use of antimicrobials for animal growth (60.8%).	Geta and Kibret, 2021 [20].
11	Ethiopia	Mar–May 2019	Self-administered/interviewer-administered questionnaire	Urban Community dwellers	374	-59.4% of respondents had heard the term AMR. Sources of information are HCW—144 (64.8%), mass media—81 (36.5%), and friends—67 (30.2%). -51.9% believed that AMR can be reduced by rational use of antibiotics. -47.6% understood AMR risk factors to include inappropriate use of antimicrobials in terms of overuse, underuse, and failure to complete the full course of therapy.	Mengesha et al., 2020 [21].
12	Ethiopia	Jun–Jul 2021	Interviewer-administered questionnaire	Urban Community Dwellers, excluding HCW, severely ill, etc.	407	-39.8% were aware of AMR. -70.8% knew that sharing of antibiotics can cause AMR.	Simegn and Moges, 2022 [22].
13	Ethiopia	Jun–Aug 2020	Self-administered questionnaire	Urban Healthcare professionals (nurses, pharmacists, medicine, laboratory)	412	-84.7% had good AMR knowledge. -Only 17.2% have had training on AMR. -Work experience, working hours per week, work stress, knowledge of over-the-counter drugs, use of antibiotics, and self-medication practice were associated with knowledge of AMR.	Simegn et al., 2022 [23].
14	Ethiopia	Jun–Jul 2019	Self-administered questionnaire	Urban Health Sciences students	232	-86% knew that irrational use of antibiotics can lead to AMR.	Fetensa et al., 2020 [24].
15	Ethiopia	Mar 2017	Self-administered questionnaire	Urban HCWs (Physicians, nurses, pharmacists)	132	-74.3% of physicians, 47.7% of nurses, and 90.9% of pharmacists had recent information on AMR. -Regarding training, 74.3% of the physicians, 84.4% of the nurses, and 72.7% of the pharmacists responded that they did not attend training regarding AMR. -Overall, more than 90% of the practitioners consider inappropriate use of antimicrobials, poor infection control in the hospital, sub-standard qualities of antibiotics, and patients’ poor adherence as factors that promote AMR.	Gebrehiwot and Tadiwos, 2022 [25].
16	Ethiopia	Nov–Dec 2020	Self-administered questionnaire	Urban (Hospitalized patients) Patients in a public hospital	233	-69.8% had heard the term AMR; 53% agreed that AMR is a global problem -Poor knowledge of impact of AMR; 88% did not know impact of AMR. -Only 40% of respondents knew that inappropriate use of antibiotics can cause AMR.	Geta and Kibret, 2022 [26].
17	Ethiopia		Interviewer-administered questionnaire	Urban Animal Producers/Farmers (cattle, sheep, goat, and poultry)	571	-34% of the animal producers were not aware that AMU in animal production can aggravate AMR. -41% of the participants know that their imprudent use of antimicrobials in animal production can lead to AMR. -78% agreed that public awareness creation can reduce AMR.	Gebeyehu et al., 2021 [27].
18	Ethiopia	Jun–Jul 2015	Self-administered questionnaire	Urban University students	670	-Only 14.8% had adequate knowledge of AMR. -Rural residents were significantly associated with drug resistance as compared to urban residents.	Zelellw and Bizuayehu 2016 [28].
19	Ethiopia	Dec–Mar 2016	Self-administered questionnaire	Urban Final-year Paramedical Students	323	-55% had poor knowledge of AMR. -96% perceived AMR as a catastrophic and preventable public problem. -There was a statistically significant knowledge difference across departments. -Knowledge of strategies to control AMR was generally poor at 19–51% correctness in the four test questions.	Seid and Hussen, 2018 [29].
20	Ethiopia	Oct–Nov 2015	Self-administered questionnaire	Urban Paramedical staffs	218	-Overall, 62.8% of paramedical staff had good knowledge of the factors causing AMR including poor adherence (96.5%), self-medication practice (96.5%), and empiric antibiotic use (94.5%). -There was significant variation in knowledge of AMR among participants, being highest among pharmacists (83.9%) and lowest among midwives (38.1%).	Tafa et al., 2017 [30].
21	Ethiopia		Self-administered questionnaire	Rural/Non-Urban Dwellers in a rural area (staff in community drug retail outlets)	276	-76% demonstrated good knowledge of AMR -58% dispense antibiotics without prescription -Noted contributors to AMR as inappropriate use of antibiotics (81.2%), dispensing without prescription (77.5%), incomplete antibiotic course (82.6%), and clients’ self-medication with antibiotics (74.6%).	Belachew et al., 2022 [31].
22	Ethiopia	Apr–Jul 2021	Interview administered questionnaire	Urban Residents	400	Only 35% have high knowledge of AMR; 17% have low knowledge.	Dejene et al., 2022 [32].
23	Gabon	Feb–Jun 2020	Self-administered questionnaire	Urban Physicians and Nurses	47	-64% noted AMR as a national problem while only 30% AMR noted it as a problem in their local hospitals. -Causes of antimicrobial resistance were recognised as excessive use of antibiotics without laboratory guidance (79%) and non-prescription use of antibiotics (79%). -Knowledge of AMR was significantly higher among physicians compared to nurses.	Adegbite et al., 2022 [33].
24	Gambia	2016	Self-administered questionnaire	Urban Health care workers (nurses-63.3%, pharmacists-6%, physicians-5.8% etc)	225	-88.24% saw AMR as a national problem. -90.37% AMR caused by abuse of antibiotics.	Sanneh et al., 2020 [34].
25	Ghana	Jan–Mar 2014	Self-administered questionnaire/in-depth interviews	Urban Prescribers (nurses-188,50%; physician assistants-69, 19%, etc.) Health care workers	379	-81.8% agreed that antibiotics currently in use may not be effective in the future (i.e., AMR), with more doctors in agreement than CHOs (96.1% vs. 69.0%). -No single formal source of information on AMR.	Asante et al., 2017 [35].
26	Ghana	May–Sep 2023	Multi-country survey Self-administered questionnaires	Human healthcare professionals	106	-Respondents had mean antibiotic resistance awareness score of 61.2%. -Antibiotic resistance awareness scores were significantly different across professions with mean scores of pharmacists (68.7%) and dentists (71.4%) higher than that of doctors (59.7%).	Pinto Jinenez et al., 2023 [36]^b^.
27	Ghana	Aug 2015	Self-administered questionnaire	Urban Physicians in a tertiary health facility	159	-30.1% of respondents perceived AMR as an important global problem, 18.5% as a national problem and 8.9% as a problem in their hospital, while only 5.5% as a problem in their department	Labi et al., 2018 [37].
28	Ghana	Aug–Nov 2019	Self-administered questionnaire	Urban Community Dwellers	632	-75.9% knowledge of bacterial ability to become resistant to antibiotics. -34.8% AMR transmissible from person to person and 34.8% from animals to humans.	Effah et al., 2020 [38].
29	Ghana		Self-administered questionnaire	Urban Meat consumers in a metropolis	384	-55% heard of AMR from teachers/school. -64% AMR occurs in germs. -49% AMR infections are difficult to treat.	Ananchinaba et al., 2022 [39].
30	Ghana	Jun-Oct 2021	Self-administered web-based questionnaire	Urban Healthcare Students (medicine, pharm and nursing)	160	-Healthcare students in higher levels (5th year) had better knowledge about AMR than those in lower years of study. -pharm/medic also better than nursing/allied.	Sefah et al., 2022 [40].
31	Ghana	Jul–Sep 2021	Self-administered questionnaire	Urban Out-patient Health seekers in Tertiary hospitals	800	-Less than 40% of respondents knew about AMR. -59% knew that AMR could prolong hospital stay, 74% knew that it could affect mortality.	Otieku et al., 2023 [41].
32	Kenya	Apr–Oct 2019	Self-administered questionnaire	Urban Prescribers (Clinical officers, Medical Officers, Pharmacists)	240	-AMR is known to be a problem worldwide (96.3%) and in the country (92.1%), but 71.6%, *p* = 0.013 agreed AMR is a problem in their HCF; near absence of antibiogram with diverse sources of knowledge on AMR but outside training institutions. -80% agreed that AMR is caused by overuse of antibiotics driven by patient demand (67.5%) and over-the-counter sales (94.6%).	Kamita et al., 2022 [42].
33	Kenya	Sep–Nov 2015	Self-administered questionnaire	Urban Physician only	107	-97.2% knew AMR to be a worldwide problem, while 93.4% knew it to be a problem locally. -75.9% noted AMR as a problem in daily practice.	Genga et al., 2017 [43].
34	Kenya	Oct–Nov 2018	Survey in 3 East African countries Self-administered questionnaire	Urban Final year Healthcare {medical and pharmacy}c students in 3 universities	75	-65% had good knowledge AMR. -97.6% had knowledge that inappropriate use of antibiotics can lead to resistance.	Lubwama et al., 2021 [44]^c^.
35	Liberia	Jul–Aug 2022	Self-administered questionnaire	Urban Healthcare professionals {Physicians, Pharmacists and Nurses}	126	-86% of physicians, 81% of pharmacists and 61.7% of nurses disagree that AMR not an issue in the country. -37.9%, 43.8% and 32.1% (physicians, pharmacists and nurses) agreed that bacteria that are resistant to antibiotics could be spread from person to person.	Paye and McClain 2022 [45].
36	Malawi	Jul-Nov 2022	Self-administered questionnaire	Urban Veterinary drug dispensers	68	-76.5% were aware of AMR and its occurrence in livestock and humans. -67.7% knew that careless use of drugs contributed to AMR in livestock.	Kainga et al., 2023 [46].
37	Malawi	February 2016	Self-administered questionnaire	Urban Final-year Medical Students	74	-83.7% believed that AMR is not a problem at the hospital level, while 86.1% believed that it is a national problem. -79.2% knew that better use of antibiotics can reduce AMR.	Kamoto et al., 2020 [47].
38	Nigeria	Aug–Sep 2022	Self-administered questionnaire	Urban Patients (out-patients)	400	-17% (68) had good knowledge of AMR, 49.3% (197) had poor knowledge. -There was a significant association between respondents’ age, marital status, level of education, and level of knowledge of AMR.	Idoko et al., 2023 [48].
39	Nigeria	Apr 2018	Self-administered questionnaire	Urban Medical students	184	-64.7% (119) had good knowledge of AMR. -AMR knowledge was associated with respondent’s gender (*p* = 0.035).	Okedo-Alex et al., 2019 [49].
40	Nigeria	Aug–Sep 2014	Self-administered questionnaire	Urban Patent medicine vendors	197	-87.3% were aware of AMR. -Had good knowledge of causes (94.9%) and prevention (98%) of AMR. -Perceived AMR as public threat (89.4–95.4%). -59.9% dispense antibiotics without prescription. -49.2% practice self-medication.	Awosan et al., 2019 [50].
41.	Nigeria	Nov 2019–Feb 2020	Self-administered questionnaire	Urban Healthcare students (Pharmacy, Dentistry, Medicine, Nursing, and Medical Laboratory Science)	576	-77.9% of students had good knowledge of AMR. -More than 60% know the common drivers of AMR.	Bello et al., 2021 [51].
42	Nigeria	Sep–Oct 2015	Multi-country survey Face-to-face interviewer-administered questionnaire	Multi-country awareness survey in 12 countries involving the public	664	-Only 38% had heard of antibiotic resistance; among them, 81% knew what it implies. -Only 57% knew that AMR is a global problem. -64% knew that antibiotic-resistant infections are increasing.	WHO, 2015 [52]^d^.
43	Nigeria		Interviewer-administered questionnaire	Semi-urban Farmers (cattle, fish and poultry) and veterinary drug shop owners	150	-50% knew the term AMR. -62% believed that AMR is other countries’ problem. -Majority did not know that AMR could be spread from human to human (58%). -Poor knowledge of causes of AMR Including indiscriminate use in animals (53.3%), and suboptimal dosing of antimicrobials in animals (53.3%).	Oyebanji and Oyebisi, 2018 [52].
44	Nigeria	May–Sep 2023	Multi-country survey Self-administered questionnaires	Human healthcare professionals	112	-Respondents had mean antibiotic resistance awareness score of 59.7%. -Antibiotic resistance awareness scores were significantly different across professions with mean scores of pharmacists (62.4%) and doctors (59.3%) higher than that of dentists (54.%).	Pinto Jinenez et al., 2023 [36]^b^.
45	Nigeria	Mar–Apr 2018	Self-administered questionnaire	Urban Breastfeeding mothers in public hospital	321	-43.7% had not heard of the term AMR. -74.6% did not know what AMR entails. -51.3% had knowledge of how AMR spread; 24% did not.	Salihu Dadari, 2020 [53].
46	Nigeria	Aug–Sep 2018	Self-administered questionnaire	Urban Veterinary students in 10 universities across 6 geopolitical zones	426	-60% demonstrated poor knowledge of AMR. -33.2% had poor knowledge of contributory factors to AMR. -Proportion with good knowledge of AMR increased relative to the year of study. -Students (50.0%) between 22 and 26 years were four times more likely to have good overall knowledge of AMR (*p* < 0.001) than other age categories.	Odetokun et al., 2019 [54].
47	Nigeria	Aug 2016–Apr 2017	Interviewer-administered questionnaire	Semi-urban Poultry farmers	152	-63% perceived that inappropriate use causes emergence of bacteria resistance; 25% did not. -65.8% expressed that AMR in broiler chickens is not a public health concern. -67.1% believed that increase in frequency of antimicrobial use cannot cause AMR in future.	Oloso et al., 2022 [55].
48	Nigeria	Feb–Mar 2021	Self-administered online questionnaire	Medical laboratory scientists across HCFs	117	-65.2% had good knowledge related to AMR, 34.8% had poor knowledge. -76% reported that AMR is a problem in their establishment. -Only 30% of establishments provided formal training on resistance testing while 66% did not have this training.	Huang and Eze, 2023 [56].
49	Nigeria	Apr–Jun 2019	Self-administered questionnaire	Urban Physicians in 6 tertiary healthcare facilities in 4 geopolitical regions in the country.	323	-82.7% had good AMR knowledge. -AMR was recognized as a global and local problem by 95.4% and 81.1% of respondents, respectively.	Babalola et al., 2020 [57].
50	Nigeria	Jun–Aug 2017	Self-administered questionnaire	Urban Veterinary Students in 5 out of 10 registered universities offering Veterinary Medicine in Nigeria	95	-72% knew that AMR is a global problem. -9% believed that AMR is not a significant problem in the country. -55% knew that AMR is promoted by overuse of antibiotics and 8% knew that poor infection control practices contribute to AMR.	Anyanwu et al., 2018 [58].
51	Nigeria	Jul–Aug 2017	Self-administered/interviewer-administered questionnaire	Urban Undergraduate students and community members	1230	-Undergraduate students displayed less knowledge that self-medication could lead to AMR than other community members (32.6% vs. 42.2%).	Ajibola et al., 2018 [59].
52	Nigeria	Jul–Sep 2021	Self-administered questionnaire	Urban Final-year undergraduate Pharmacy students	164	-94.5% aware of antimicrobial resistance. -Knowledge about contributors to AMR among respondents includes poor adherence (86.6%), overuse of antimicrobials in humans (82.3%), substandard quality of antimicrobials (75%), and poor handwashing practices (39%).	Abdu-Aguye et al., 2022 [60].
53	Nigeria	2014	Self-administered questionnaire	Urban Physicians	105	-57.1% lack the up-to-date information on AMR. -81.9% had no training on AMR.	Ahmad et al., 2015 [61].
54	Nigeria	Jul–Nov 2019	Self-administered online questionnaire/self-administered questionnaire	Urban Final year medical students in two countries (Nigeria and South Africa)	172	-11% agreed that AMR is a problem in their hospitals. -93.0% knew inappropriate antibiotic use causes resistance. -84.3% of use of broad-spectrum antibiotics could cause AMR while only less than 2/3 knew that lack of hand disinfectant promotes AMR.	Augie et al., 2021 [62]^e^.
55	Nigeria	Jun–Nov 2019	Self-administered questionnaire	Urban Healthcare workers in 6 geopolitical zones in Nigeria	358	-Physicians had better knowledge of AMR than other HCWs; HCWs in the tertiary HCFs had better knowledge than those in primary and secondary HCFs. -Overall, 49.2% had good AMR Knowledge, 47.2 had fair and 3.6% had poor	Chukwu et al., 2020 [63].
56	Nigeria	Aug–Nov 2019	Self-administered questionnaire	Urban Healthcare Students	866	-58.4% had good knowledge of AMR. -students in years 3–6 had greater knowledge of AMR compared with those in years 1 and 2.	Akande-Sholabi and Ajamu 2021 [64].
57	Nigeria	Jun–Nov 2019	Self-administered/interviewer-administered questionnaire	Urban/rural Community dwellers in 6 geopolitical zones	482	-56.5% familiar with the term AMR. -Only 8.3% had good knowledge of AMR. -Significant variation in knowledge of AMR across the regions in the country.	Chukwu et al., 2020 [65].
58	Nigeria	Jan–Mar2022	Self-administered questionnaire	Urban Healthcare workers	600	-Respondents’ knowledge of AMR is 58.8%.	Nwafia et al., 2022 [66].
59	Rwanda	Mar 2017	Self–administered questionnaire	Urban Healthcare students {medical, dental and pharmacy students}	229	-Students in Levels 3 to 6 had better knowledge of AMR than those in lower levels.	Nisabwe et al., 2020 [67].
60	Senegal	Jul–Oct 2019	Self-administered questionnaire	Urban Undergraduate Pharmacy Students	278	-85.6% had good knowledge of AMR.	Bassoum et al., 2023 [68].
61	Senegal	Nov–Dec 2017	Interview-administered questionnaire	Urban People attending bus station (HCWs were excluded)	400	-Only 8.8% and 41.8% knew that handwashing and vaccination prevent AMR. -7% had good knowledge. -83.8% knew high antibiotic consumption can lead to bacterial resistance.	Bassoum et al., 2023 [69].
62	South Africa	Nov 2017–Jan 2018	A national cross-sectional survey Self-administered online questionnaire	Doctors, pharmacists and nurses in public and private employment	2523	-The majority of HCPs (93.37%) perceived AMR to be a serious problem globally; however, a much lower number of HCPs (73.77%) agreed AMR was a serious problem in their hospital or practice. -Antimicrobial resistance was considered a severe problem globally and nationally by the majority of HCPs. -Contributory to AMR were noted as overuse of antimicrobials (by 91.6% of HCPs) and non-adherence to prescriptions (by 73.3% of HCPs). -Majority of HCPs recognised measures to combat AMR as educational campaigns (91.2%), use of therapeutic guidelines (84.7%), and improved infection control (66.3%). -Only 40.1% of HCPs attended training on AMR and 81.6% requested more education and training.	Billiram et al., 2021 [70].
63	South Africa	Sep–Oct 2015	Multi-country survey Face-to-face interviewer-administered questionnaire	Multi-country awareness survey in 12 countries involving the public	1002	-77% had heard of antibiotic resistance; among them, 83% knew what it implies. -Only 55% knew that AMR is a global problem. -72% knew that antibiotic-resistant infections are increasing	WHO, 2015 [71]^d^.
64	South Africa	April 2016–May 2017	Cross-sectional study Self-administered questionnaire	Patients in public and private primary healthcare facilities	782	-62% of patients knew that AMR occurs when germs become resistant as people take too many antibiotics. -58% of patients knew that AMR is costly to remedy worldwide, the fact which was more commonly known by patients with high knowledge of AMR in private (72%) and public (80%) HCFs.	Farley et al., 2019 [72].
65	South Africa	2015	Cross-sectional study Self-administered questionnaire	Final-year medical students in three medical schools	289	-87% agreed that resistance is a significant problem in the country and 61% agreed that AMR is a problem in the hospitals where they had worked. -More than 95% of students knew that inappropriate use of antibiotics causes antibiotic resistance. -Most (90%) students reported that they would appreciate more education on antibiotic resistance.	Wasserman et al., 2017 [73].
66	South Africa	Jul–Nov 2019	Self-administered online questionnaire/self-administered questionnaire	Urban Final year medical students in two countries (Nigeria and South Africa)	104	-48% agreed that AMR is a problem in their hospital. -99% knew inappropriate antibiotic use causes resistance. -91.4% of use of broad-spectrum antibiotics could cause AMR while only less than 2/3 knew that lack of hand disinfectant promotes AMR.	Augie et al., 2021 [62]^e^.
67	South Africa	Oct 2015–Dec 2016	Cross-sectional study Self-administered questionnaire	Primary healthcare prescribers	264	-95.8% (230/240) believed that ABR is a big problem in the country. -Most of the prescribers generally had good knowledge of AMR and its driver and those with high knowledge were more likely to believe that resistance can be reduced by using narrow-spectrum antibiotics. -The majority (226/235, 96.2%) requested data on local resistance patterns, and 90.4% (208/230) requested education resource aids for discussions on AMR with patients.	Farley et al., 2018 [74].
68	South Africa	2014	Self-administered questionnaire	University undergraduate veterinary students	71	-All respondents knew that AMR is an increasing threat to humans and animals. -Driver of AMR was noted to be inappropriate antimicrobial use among veterinary practitioners by 84% of students, and among farmers by 98% of students. -55% of the students believed that AMR can be reduced with a ban on the use of antimicrobials as prophylactics and growth promoters in food animals.	Smith et al., 2019 [75].
69	Tanzania	May–Sep 2023	Multi-country survey Self-administered questionnaires	Human healthcare professionals	124	-Respondents had mean antibiotic resistance awareness score of 56.6%. -Antibiotic resistance awareness scores were significantly different across professions with mean scores of pharmacists (61.9%) and doctors (60.4%) higher than those of dentists (54.1%) and nurses (54.7%).	Pinto Jinenez et al., 2023 [36]^b^.
70	Tanzania	Sep–Nov 2019	Qualitative study using phenomenographic approach Face-to-face interview with audio recording	Prescribing healthcare workers in five health centres and seven dispensaries	20	-Many HCWs had the knowledge that limited access to antibiotics can cause antibiotic resistance. -Most healthcare workers were aware of the issue of antibiotic resistance, but few experienced it as a problem in daily practice. -Most healthcare workers perceived antibiotic resistance as a problem of individuals who misused antibiotics while few saw it public health problem.	Emgard et al., 2021 [76].
71	Tanzania	Nov–Dec 2921	Mixed method approach (quantitative and qualitative survey)	Pastoralists/livestock farmers	250	-Only 32% aware of AMR.	Mangesho et al., 2021 [77].
72	Tanzania		Interviewer-administered questionnaire	One person per household in 4 regions (12 districts) of the country	1200	-Knowledge of existence of AMR was poor across infection syndromes (22.6–38.6%). -Knowledge of drivers of AMR is also poor among respondents (41.8–45.8%). -Respondents who completed primary education were three times more likely to have more knowledge than those with no or incomplete primary education.	Simba et al., 2016 [78].
73	Tanzania	May–Jun 2019	Survey in 3 East African countries Self-administered questionnaire	Urban Final-year healthcare (medical and pharmacy) students in 3 universities	178	-Only 44% had good knowledge of AMR. -97.7% had knowledge that inappropriate use of antibiotics can lead to resistance.	Lubwama et al., 2021 [44]^c^.
74	Tanzania	Jul 2010–Jan 2011	Interviewer-administered questionnaire	Small-scale livestock keepers	160	-30% of respondents were not aware of antibiotic resistance.	Katakweba et al., 2012 [79].
75	Tanzania		Qualitative semi-structured interview	Veterinary paraprofessionals in 5 community districts	40	-Most reported that they have not attended refresher courses or seminars on AMR which has limited their understanding of AMU and AMR issues. -Reported that their clients (livestock keepers) have little understanding of AMR.	Frumence et al., 2021 [80].
76	Tanzania	Jan–Feb 2020	Community-based cross-sectional study Interviewer-administered questionnaire	Community participants in three districts	828	-Low to moderate level knowledge of AMR. -Levels of knowledge were significantly influenced by increased participant’s age and level of education,	Sindato et al., 2020 [81].
77	Togo	Aug–Sep 2019, Oct–Nov 2020	Cross-sectional study Interviewer-administered questionnaire	Commercial poultry and pig farmers	218	-39% of poultry farmers and 57% of pig farmers were unaware of antibiotic resistance. -No adequate ABR knowledge in 19% of poultry farmers and 64% of pig farmers.	Bedeke labou et al., 2022 [82].
78	Togo	Jan–Jul 2021	Survey including 2 countries in West Africa Self-administered/interviewer-administered questionnaire	Urban Health Professionals (physicians, pharmacists and veterinarians)	221	-84% had good/very good knowledge of AMR. -No difference as regards the proportions of respondents with good knowledge of AMR across the professions.	Bedekelabou et al., 2022 [16]^a^.
79	Uganda	Apr–May 2021	Cross-sectional study Self-administered questionnaire	HCWs (physicians, nurses and pharmacists) in a national cancer institute	61	-All respondents had heard of the term AMR but degree of knowledge of AMR is significantly lower among nurses compared to pharmacists or physicians. -85% of respondents agreed that AMR is a problem for patients in the HCF. -Most respondents (81–85%) respectively identified various AMR-causing practices bordering on inappropriate and excessive antibiotic uses, while only 50% knew that poor hand hygiene is an important cause of infection by AMR bug.	Gulleen et al., 2022 [83].
80	Uganda	Oct 2021	Descriptive cross-sectional, multicenter, online survey with semi-structured questionnaire	Clinical health sciences undergraduate students across 9 universities	681	-Most participants (87.5%) had sufficient knowledge of AMR. AMR knowledge significantly higher among students at higher levels and those with previous teaching on AMR.	Kayinke et al., 2022 [84].
81	Uganda	Oct–Nov 2018	Survey in 3 East African countries Self-administered questionnaire	Urban Final-year healthcare (medical and pharmacy) students in 3 universities	75	-67% had good knowledge of AMR. -96% had knowledge that inappropriate use of antibiotics can lead to resistance.	Lubwama et al., 2021 [4]^c^.
82	Uganda	Jun–Sep 2021	Cross-sectional study Interviewer-administered questionnaire	Members of farming households (crop and animal)	652	-The majority of participants were able to correctly describe antibiotics and are aware of AMR; however, there was some degree of misunderstanding of several AMR concepts. -Most (77%) respondents knew that infections are becoming increasingly resistant to treatment and difficult to treat, but only 9.2% understood what AMR implies. -83% knew that AMR can affect individuals or families but about 32% believed that it is a problem of foreign countries. 63.8% wrongly thought that AMR only affects individuals who regularly take antibiotics. -60% of respondents knew that AMR can complicate surgical procedures.	Muleme et al., 2023 [85].
83	Uganda		Cross-sectional qualitative and quantitative study Seif-administered questionnaires	Prescribing and dispensing HCWs in 4 primary healthcare facilities in rural communities	124	-75% of respondents reported receiving information about antibiotic resistance with medical training school (67.2%) being the main source of information. Only 54.8% had knowledge of drug-resistant bacteria. -The respondents with knowledge of the drivers of antibiotic resistance accounted for only 23.5%, although most of them (75.4%) knew such drugs that have been rendered ineffective in treating infections.	Amelia et al., 2017 [86].
84	Zambia	Oct 2018–Jun 2019	Self-administered questionnaire	Undergraduate medical students	260	-87.3% had good knowledge of AMR. -59.6% agreed that misuse is the leading cause of AMR. -Students at higher levels have significantly higher knowledge of AMR than those at lower levels.	Zulu et al., 2020 [87].
85	Zambia	Sep 2020–Apr 2021	Cross-sectional study Interviewer-administered questionnaire	Layers poultry farmers	77	-Overall awareness of AMR was low among poultry farmers (47%). -Awareness of AMR was more among commercial farmers, farmers who use prescriptions to access antibiotics, and those who did not use antibiotics on market-ready birds.	Mudenda et al., 2022 [9].
86	Zambia	Jan–Apr 2022	Cross-sectional study Self-administered questionnaire	Pharmacy personnel and nurses in tertiary hospital	263	-Only 54.4% of the participants knew that AMR is a public health problem while most (85.9%) knew that infections with antibiotic-resistant bacteria are difficult to treat. -Pharmacy personnel had better knowledge than nurses that resistant bacteria are spread from person to person, and that the use of antibiotics in livestock contributes to AMR.	Tembo et al., 2022 [88].
87	Zambia	Mar 2021–Mar 2022	Self-administered questionnaire	Medical students from six medical schools (first to final year)	180	-The students (96.7%) had good to excellent overall knowledge of AMR. -Clinical students had six times the likelihood to have excellent knowledge of AMR than pre-clinical students.	Nowbuth et al., 2023 [89].
88	Zambia		Cross-sectional survey Self-administered questionnaire	Healthcare professionals in tertiary hospitals (physicians, nurses, pharmacists and biomedical personnel)	304	-Pharmacists had the highest score for AMR knowledge while nurses had the lowest. -A minority of respondents indicated that poor access to local antibiogram data (31.5%) and poor IPC in hospitals (31.3%) promoted AMR, while majority (56.7%) noted that poor adherence to prescribed antimicrobials was the main cause of AMR.	Mufwambi et al., 2020 [90].
89	Zambia	Jan–Jul 2018	Cross-sectional survey Self-administered questionnaire	Undergraduate pharmacy students	172	-90% had overall knowledge of AMR while only 54.1% knew that AMR is a global problem.	Mudenda et al., 2022 [91].
90	Zambia	Feb–Apr 2022	Cross-sectional study Self-administered questionnaire	Community pharmacists and pharmacy technologists who dispense poultry drugs	178	-Most (96.6%) of the participants were aware of AMR. -The study found moderate knowledge (mean score of 64.7%). -Good knowledge of AMR was associated with work experience of more than one year.	Mudenda et al., 2022 [92].
91	Zambia	Nov–Dec 2021	Cross-sectional survey. Self-administered questionnaire	Poultry farmers	106	-29.2% were aware of AMR. The study showed that 46.2% of the participants had low knowledge of AMR.	Chilawa et al., 2023 [93].
92	Zimbabwe	Oct–Dec 2020	Cross-sectional survey Self-administered questionnaire	Low-income suburbs Nurse-led healthcare providers in 9 primary health out-patient clinics	91	-AMR was considered a global problem (82%), a national problem (89%), and an HCF problem (57%). -They have good knowledge of some drivers of AMR, including poor adherence to prescription and excessive unregulated access to antibiotics, as well as poor knowledge of other drivers including substandard drug quality and poor IPC.	Olaru et al., 2023 [94].

^a^ Multi-country studies (Cote d’ Ivoire and Togo); ^b^ multi-country studies (Ghana, Nigeria, Tanzania); ^c^ multi-country studies (Kenya, Tanzania, Uganda); ^d^ multi-country studies (Nigeria and South Africa); ^e^ multi-country studies (Nigeria and South Africa); AMR—antimicrobial resistance; AMU—antimicrobial use; HCF—healthcare facility; HCP—healthcare practitioner; HCW—healthcare worker; IPC—infection prevention and control.

## Data Availability

The information presented in this review is available in open access journals, as indicated in Table 1 and under references.

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
