# Peer review of "Education and Awareness on Antimicrobial Resistance in the WHO African Region: A Systematic Review"

_antibiotics, 2023, doi:10.3390/antibiotics12111613_

Round 1

Reviewer 1 Report

Minor English editing needed i.e. line 12, line 40.

I think the authors could have opted for more keywords and manuscript would have more merit were there 2 independent researchers. It is a pitty the researches did not try to obtain all eligible manuscripts in full version and just chose to exclude them. It is unclear whether manuscripts that included studied who region only in a part were included - please clarify. I believe that the final number of manuscripts is suprisingly low for such interesting topic.

Table 1 needs to be added as supplementary file and results presented differently and in greater detail

PRISMA guidelines are mentioned only in the Abstract. It is unclear whether the quality of the included manuscripts was evaluated according to PRISMA? Please clarify the relevance of PRISMA in this study.

Minor English editing needed i.e. line 12, line 40. 

Reviewer 2 Report

Fuller et al report on a systematic review on AMR-related education awareness and implementation initiatives in Africa, in line with the objective of the Global Action Plan on AMR – “Improve knowledge and 12 understanding on AMR through effective communication, education and training”. The articles (48) included in the analysis were selected from a pool of 248,157 potentially eligible studies. The authors’ findings suggest concomitant low AMR knowledge and wide antimicrobial misuse in Africa.

The study of Fuller et al has a high public health significance, given the importance of knowledge and perceptions in shaping practices. The population chosen for the evaluation was strategic, and increases the importance of the study – in the 2019 Global AMR Burden study, the various major geographical zones of Africa accounted for the highest burden. It appears that while major efforts have been made concerning AMR surveillance in the region, robust evaluation of AMR knowledge, perceptions, and practices have received relatively less attention – that is another lesson from Fuller et al’s report, which could guide antimicrobial stewardship in the region.

The manuscript is well written, the presentation is clear, and the conclusions are in line with the results and meet the objectives of the study. The authors also provided sufficient background that highlight the importance of their study. Additionally, the methods are adequately described, and the results clearly presented and discussed. The references cited are also appropriate.

There are a few issues that the authors may want to correct, mainly with a needed clarification in the Methods and some grammatical issues, outlined as follows:

1.      The authors may want use Medline or PubMed, or better still, clarify the relationship between Medline and PubMed in their explanation of the databases they searched for potentially eligible studies.

2.      Although authors report including a total of 48 studies in their analysis, in Table 1, they have outline 49 studies. This needs to be clarified.

3.      Issues with Grammar

a.      Lines 11, 70, 138, 142, and 223, and Table 1 (Studies 11 and 120: The context in which “region” is used demands that it be written as a proper noun

b.      Lines 14, 68, 82, 92, and 109: “Google scholar”, “google scholar” need to be written as proper nouns, that is “Google Scholar”.

c.       Line 29: The authors may want to substitute “Antimicrobial resistance” with its pronoun (It) to make the sentence more coherent with the previous sentence.

d.     Lines 54 and 55: Please replace the semicolons with commas.

e.      Line 74: Please write “table 1” as “Table 1”.

f.        Line 83: Please introduce a comma after “countries”.

g.      Line 88: Please introduce a full stop after “review”.

h.      Line 90: The authors may want to put the figure caption below the figure.

i.        Line 109: The sentence is not clear.

j.        Line 113: The statement “In almost of the articles” is not clear.

k.      Lines 149 to 153: The sentence is too long; it needs to be broken into shorter ones.

l.        Line 155: Please rewrite “investigate on” as ““investigate”.

m.   Table 1: The key finding stated for Study 1 (done in Angola) is not clear, and the “et. Al.” needs to be “et. al.”.

Moderate English editing required.

Round 2

Reviewer 1 Report

Authors have made substantial changes to the Manuscript